# The Emerging Role of Hypoxic Training for the Equine Athlete

**DOI:** 10.3390/ani13172799

**Published:** 2023-09-03

**Authors:** Allan Davie, Rosalind Beavers, Kristýna Hargitaiová, Joshua Denham

**Affiliations:** 1Australian Equine Racing and Research Centre, Ballina, NSW 2478, Australia; 2Faculty of Health, Southern Cross University, Lismore, NSW 2480, Australia; rosalind.beavers@scu.edu.au; 3Department of Clinical Sciences, Cornell University, 930 Campus Rd, Ithaca, NY 14850, USA; kristyna.hargitaiova@gmail.com; 4School of Health and Medical Sciences, University of Southern Queensland, Toowoomba, QLD 4305, Australia; josh.denham@usq.edu.au; 5Centre for Health Research, Toowoomba, QLD 4350, Australia

**Keywords:** horse, hypoxia, mitochondria, hypoxia-inducible factor, equine, normobaric, thoroughbred

## Abstract

**Simple Summary:**

The concept of altitude training became popular among human athletes following the 1968 Olympic Games, at which African runners were particularly successful. Culminating from these observations was the concept that during exercise training, local tissue hypoxia is an important adaptive stress for muscle that ultimately leads to superior physiological adaptations and enhanced endurance performance. The application of the concept of hypoxic training to the Thoroughbred horse is new, and now, with purpose-built hypoxic chambers, there has been a growing interest in its use in equine training programs.

**Abstract:**

This paper provides a comprehensive discussion on the physiological impacts of hypoxic training, its benefits to endurance performance, and a rationale for utilizing it to improve performance in the equine athlete. All exercise-induced training adaptations are governed by genetics. Exercise prescriptions can be tailored to elicit the desired physiological adaptations. Although the application of hypoxic stimuli on its own is not ideal to promote favorable molecular responses, exercise training under hypoxic conditions provides an optimal environment for maximizing physiological adaptations to enhance endurance performance. The combination of exercise training and hypoxia increases the activity of the hypoxia-inducible factor (HIF) pathway compared to training under normoxic conditions. Hypoxia-inducible factor-1 alpha (HIF-1α) is known as a master regulator of the expression of genes since over 100 genes are responsive to HIF-1α. For instance, HIF-1-inducible genes include those critical to erythropoiesis, angiogenesis, glucose metabolism, mitochondrial biogenesis, and glucose transport, all of which are intergral in physiological adaptations for endurance performance. Further, hypoxic training could conceivably have a role in equine rehabilitation when high-impact training is contraindicated but a quality training stimulus is desired. This is achievable through purpose-built equine motorized treadmills inside commercial hypoxic chambers.

## 1. Introduction

The performance of the equine athlete consists of several variables, such as environmental, physiological, genetic, and training regimens. For the trainer, the major contribution to performance outcomes is the process of exercise training, particularly its structure and content. Hypoxic training involves exercising in a reduced oxygen environment. Oxygen is an important metabolite critical for maintaining a stable cellular and physiological environment and organism survival. Hypoxic training is a popular training strategy amongst human endurance athletes, as exercising or living in hypoxia improves muscle function through hypoxia-inducible factor 1α (HIF-1α)-mediated gene signaling and is linked to performance benefits, especially in endurance-based competitions. Furthermore, temporary hypoxic states may not only improve athletic performance but also provide potential health benefits. For example, hypoxia interventions are linked to key cellular events in normal cell development and in pathological settings such as ischemia [1].

Considering that coaches and human athletes have been employing hypoxic training to gain competitive advantages for many years and that it is a topical and emerging area in racehorses, it is timely to provide a discussion on the current understanding in relation to equine athletic performance. This is necessary because often assumptions are made based on human research, and the findings in one species may not always translate to the findings in another. Therefore, the purpose of this commentary is to address specific factors governing equine performance and the adaptations provided by hypoxic training that have the potential to improve performance in the equine athlete.

## 2. Factors Governing Performance

Performance is a complex outcome involving a dynamic network of biological (genetics—nature), environmental (nurture), and biomechanical variables; here, we focus on biological variables. In humans, research has linked several genes to athletic performance, most notably insulin-like growth factor 1 (*IGF1R*), peroxisome proliferator-activated receptor γ coactivator 1α (*PPARGC1A*), alpha-actinin-3 (*ACTN3*), and monocarboxylate transporter 1 (*MCT1*) [2]. In the equine field, single nucleotide polymorphisms (SNPs) in creatine kinase (CKM) and cytochrome c oxidase (COX412) have been linked to elite Thoroughbreds; the myostatin (MSTN) gene has been shown to influence early skeletal muscle development and aptitude for sprint racing; and the mitochondrial haplogroup L3b has been shown to have a negative association with elite Thoroughbreds [3,4,5]. This list is expected to expand significantly in the future since whole-genome sequencing technology has improved and the cost has been reduced. Today, artificial selection of the most talented and successful racehorses, with genetic screening for optimal genetic polymorphisms, remains the current best practice for attempting to preselect performance. Conversely, environmental factors are critical to a successful athlete, none more so than exercise training. Therefore, we will limit our discussion to exercise prescription.

From a physiological perspective, maximum oxygen uptake ($\dot{V}$O_2max_) is arguably the most important factor limiting endurance performance in both humans and horses [6,7]. $\dot{V}$O_2max_ is determined by cardiac output (stroke volume × heart rate) and arterial-venous oxygen difference (a-v_O2_ Diff). $\dot{V}$O_2max_ is primarily limited by two factors: cardiac output, or the capacity for delivering blood (oxygen) to muscles, and the capacity of muscles to utilize oxygen to transfer energy, which is dependent on mitochondrial density and function. In humans, $\dot{V}$O_2max_ has been used as an indicator for endurance athletes’ potential success. Although other physiological attributes are critical to performance in humans (e.g., lactate inflection point), a high $\dot{V}$O_2max_ is an indisputable requirement at the elite level. This, however, has not been demonstrated in horses, since the $\dot{V}$O_2max_ of Thoroughbred horses can vary from ~100 to 160 mL·kg^−1^·min^−1^ [7,8,9]. Considering the physiological profiles of elite racehorses are not commonly reported in the literature, the performance outcomes of horses with superior $\dot{V}$O_2max_ remain to be established.

In horses, ref. [7], $\dot{V}$O_2max_ continuously increases with training over several weeks to months, as shown in a study by Tyler et al. [7]. They examined the relationship between peripheral (skeletal muscle) and whole body ($\dot{V}$O_2max_) adaptations to training and reported significant adaptations in the morphological characteristics of skeletal muscle (increases in fiber area and capillarization), which were limited and largely completed by 16 weeks of training. In contrast, mitochondrial volume continued to increase throughout the 34 weeks of training and paralleled the increases in $\dot{V}$O_2max_. These findings highlight that the markers of oxidative capacity in muscle progressively increase in parallel with aerobic capacity in response to increases in training load. This concept is supported in human studies [10], which report that although oxygen delivery is the main limiting factor for $\dot{V}$O_2max_, enhanced oxygen extraction fraction contributes to the remarkably high $\dot{V}$O_2max_ in endurance-trained individuals. To quantify mitochondrial oxygen consumption directly requires specialized equipment and invasive procedures, which makes it challenging to investigate the relative importance of central versus peripheral adaptation for equine performance. It is tempting to speculate that maximum mitochondrial oxygen consumption and muscle glycolytic function may limit short-distance (i.e., <5 km) flat race performance, whereas lactate inflection point and running economy might be bigger predictors of endurance races (e.g., >60 km).

Within skeletal muscle, the primary sites of energy production are the cytoplasm for anaerobic (glycolysis) and the mitochondria for aerobic (oxidative phosphorylation) metabolism. For the horse, however, even in high-intensity exercise, the energy is provided principally from the aerobic system and a smaller fraction from the ATP-PC glycolysis network. For example, in a horse performing at between 105 and 125% $\dot{V}$O_2max_ the mean energy contributions are 81–72 (aerobic) and 18–27% (anaerobic), respectively [11]. During aerobic work, the energy predominantly comes from the Krebs Cycle within the mitochondria. The mitochondria are frequently referred to as the “powerhouses” of energy production, as all aerobic energy production takes place within the mitochondria. Approximately 80% of the oxygen taken up by the cells is consumed by the mitochondria to meet metabolic demands. Therefore, development of the mitochondria is a key component of training. In humans, an adaptation appears to be fiber-specific; however, most changes result predominately from endurance-based training performed below the lactate inflection point (or anaerobic threshold). It has been proposed that the magnitude of improvement in the mitochondria is influenced by the duration of the exercise or training session [12,13,14,15]. There is, however, not a direct linear relationship, and with additional training, this factor appears to become less important. Moreover, it was also suggested that the best adaptations occur when intensity and duration of training interact. In horses, however, [7] $\dot{V}$O_2max_ continuously increases with training, with significant increases in mitochondrial volume density also occurring in parallel with increases in $\dot{V}$O_2,max_. These findings highlight that markers of the oxidative capacity of muscle progressively increase with aerobic capacity in response to increases in training load.

It is well documented that the application of a stimulus or stress to the body, such as a bout of exercise, results in an acute response followed by recovery and subsequent performance improvement. In an ideal scenario, physiological adaptations would continue to occur in response to repeated training stimuli until genetics imposed a ceiling effect. The adaptation to training represents the cumulative effect of repeated bouts of exercise and is highly specific to the exercise mode, intensity, duration, and frequency of the stimuli [16,17]. At a molecular level, adaptations to training are thought to be due to the cumulative effects of transient changes in mRNA transcripts encoding proteins that follow each acute training session [18,19]. The controlling mechanisms underlying this process are incompletely understood. The transient changes in transcription and expression of mRNA for metabolic genes have been examined following a single bout or session of exercise [20,21] and the effects on the concentration of the proteins [17,22,23]. The role of exercise-induced epigenetic modifications in exercise adaptations is also beginning to emerge in the equine field [24]. In addition to the training stimuli, a stimulus in the form of hypoxia may well provide additional training stress and greater physiological adaptations.

## 3. Hypoxic Training Is a Potent Training Stimulus

The concept of altitude training became popular amongst human athletes following the 1968 Olympic Games, at which the African runners were particularly successful and were known to have trained and lived at altitude. Culminating from these observations was the concept that local tissue hypoxia is an important adaptive stress for muscle tissue in exercise training. The fundamental concept behind hypoxic training is that the concentration of oxygen in the inspired air is reduced from 20.9% to reduced levels (e.g., 15%) with a normal atmospheric pressure (normobaric). This reduced concentration means less oxygen enters the bloodstream, resulting in reduced availability for the working muscles. Living or training in a hypoxic environment ultimately unbalances the oxygen supply and demand relationship. Due to an inadequate supply of oxygen to the cells, senses detect the oxygen deficiency and activate molecular signaling pathways to correct the imbalance. The short-term response is generally physiologically based and includes changes in respiration, heart rate, blood volume, and vasodilation. Long-term changes are achieved more at the molecular level, with intermittent exposure to hypoxia primarily resulting in an up-regulation of the regulatory subunit of hypoxia-inducible factor-1 alpha (HIF-1α) [25,26,27]. The HIF-1 protein is composed of two subunits, HIF-1α and HIF-1β. Generally, under normoxic conditions, HIF-1α is degraded through hydroxylation and binds to the Von Hippel Lindau (VHL) protein, resulting in proteasomal degradation of HIF-1α. However, under hypoxic conditions, this process is blocked, thus allowing HIF-1α to accumulate and bind to the subunit HIF-1β, which forms HIF-1. HIF-1 is a transcription factor that interacts with hypoxia response elements located in the nucleus of the cell, triggering transcription of target genes. HIF–1 has been referred to as a master regulator of the expression of genes, with more than 100 genes identified as responsive to HIF-1. For instance, HIF-1-inducible genes include those critical to erythropoiesis, metabolism, angiogenesis, glucose metabolism, mitochondria, glucose transport, and cell proliferation [26].

Importantly, the application of hypoxic stimuli alone may not elicit optimal responses for changes in gene response related to performance. The combination effect of training and hypoxia on the activity of the HIF-1 pathway has been reported to be higher than that under normoxic conditions, indicating that combining hypoxia with exercise training provides a superior stimulus. Exercise intensity provides an additional stimulus due to the increased levels of reactive oxygen species (ROS) produced during metabolic activity. The primary source of ROS production is the mitochondrial electron transport chain, which reduces the majority of oxygen to water, however, a small volume of the oxygen is converted to ROS (e.g., hydrogen peroxide or superoxide). The ROS has a role in inducing the expression of peroxisome proliferator-activated receptor-γ coactivator 1α (PGC-1α), which is a master regulator of mitochondrial biogenesis, and also in influencing increases in vascular endothelial growth factor (VEGF), a key regulator of angiogenesis [28]. Regulation of both PGC-1α and VEGF is crucial to adaptation to exercise [26,29]. However, sustained exposure to severe hypoxia has detrimental effects on skeletal muscle function, including decreases in muscle oxidative capacity and loss of muscle mass [30]. Additionally, living and training at high altitude (hypobaric hypoxia) is associated with side effects such as mountain sickness, decreased maximum heart rate and plasma volume, and impaired exercise performance in humans [31]. Conversely, short-term exposure to altitude leads to positive skeletal muscle adaptations (e.g., muscle capillary growth and mitochondrial biogenesis) [25,26,32], supporting the need to ensure that a correct balance is utilized between time spent in hypoxia and training intensity in horses.

Since the early 2000s, evidence from trials has accumulated and provided insight on the influence of various types of altitude/hypoxic training protocols, such as “living high training low” (LHTL), “living low training high” (LLTH), and “living high training high” (LHTH), on metabolic, biochemical, and other molecular adaptations. These studies were designed to establish the most ideal protocol for performance enhancement [27,33,34,35,36,37,38,39]. Despite some equivocal findings on physiological and cellular responses to hypoxic training among studies utilizing different training protocols, important practical applications have been suggested. Accordingly, the following sections provide a concise synthesis of outcomes from two of the most popular training methods, LHTL and LLTH, which have the most practical applications that could be adopted and guide hypoxic training in Thoroughbred horses.

## 4. Living High Training Low (LHTL)

A seminal study by Stray-Gunderson et al. [38] examined elite male and female runners after 27 days of living at 2500 m who concurrently performed high-intensity training at 1250 m altitude. Despite only a 3% improvement in $\dot{V}$O_2max_, increases in erythropoietin levels and running performance were noted (8.5 ± 0.5–16.2 ± 1.0 IU/mL) and performance 1.1%, respectively. Several other groups reported mixed results regarding changes in $\dot{V}$O_2max_ after training. One study [36] examined three groups of elite athletes: cross-country skiers, swimmers, and runners. Within each athlete cohort, the participants were assigned to two sub-groups: LHTL and control cohorts. Training consisted of 13–18 days at 1200 m (swimmers only had 13 days). Athletes were divided into two groups: a control group that slept at 1200 m and a hypoxic group that slept at three different altitudes: 2500 m (5 nights for swimmers and 6 for skiers and runners); 3000 m (6 nights for skiers, 8 for swimmers, and 12 for runners); and 3500 m (6 nights for skiers). Mixed results were reported for changes in erythropoiesis, with 6 days at 2500 m having little effect, whereas for 6 days at 3000 m, values were higher than basal levels. However, no significant increases in $\dot{V}$O_2max_ were observed in swimmers after 13 days. Conversely, a significant change was reached after 18 days in runners. In another study [40], a group of male and female runners who trained for three weeks in hypoxia with train high (TH) resided at sea level, while LHTL plus train high (LH/TL + TH) stayed in normobaric hypoxia for 14 h per day. All athletes completed 4–5 h of hypoxic training per week, training at ~82% maximum heart rate. They reported that the combination of LH/TL + TH provided a greater increase in $\dot{V}$O_2max_, hemoglobin mass, and 3-km time trial performance than the TH group.

Although the research highlights that the LHTL method provides some beneficial effects on performance, the logistics and associated costs make it inaccessible for many athletes [41]. Intermittent hypoxic exposure (IHE) has been proposed as an alternative form of training that involves acute 60–90 min daily exposure to hypoxia. In a study involving triathletes, participants were allocated to either a LHTL, IHE, or placebo group. The LHTL group spent approximately 14 h per day for 17 days in normoxic hypoxia (3000 m). In the IHE group, participants completed a daily period of breathing hypoxic air (concentration changed over the 17 days from 3500 to 6000 m) for 6 min and normal air for 4 min; this cycle repeated six times for a total of 60 min. There was an increase in hemoglobin mass and a 2.8% improvement in running economy following LHTL compared to placebo, yet no significant changes in $\dot{V}$O_2max_ were observed in any group [41]. These findings are supported by others. For example, in another trial, 20 endurance athletes were either exposed for 15 consecutive days to breathing either a gas mixture (11% O_2_ days 1–7 and 10% O_2_ days 8–15) or formed part of a control group [42]. Participants repeated the cycle of 6 min of hypoxic breathing with 4 min of normal air for a total of six consecutive cycles. They reported no effect from the treatment on aerobic or anaerobic performance as assessed by peak power and $\dot{V}$O_2max_ tests, respectively. Together, these studies support the premise that the application of hypoxic stimuli on its own does not appear to maximize exercise performance. Conversely, a combination of training/hypoxia seems more conducive for physiological adaptations to occur, especially a combination of LHTL or LHTL + TH.

## 5. Living Low Training High (LLTH)

The LLTH method is a convenient approach that improves endurance performance, with the majority of studies showing that muscle adaptations are superior following LLTH than those after normoxic training. This method also removes the risk of side effects from living at altitude, as sustained exposure to severe hypoxia can have detrimental effects on skeletal muscle function (loss of muscle mass and oxidative capacity) [30]. A pioneering study examined 30 untrained men who were trained on a cycle ergometer for 30 min per day, five days a week, for 6 weeks [39]. The four groups consisted of two normal low-altitude groups that trained at high (80%) and low (67%) $\dot{V}$O_2max_ and two high-altitude (3850 m) groups that trained at identical intensities. They found significant increases in $\dot{V}$O_2max_ for all groups; the mean increase in $\dot{V}$O_2max_ appeared higher in hypoxic groups, albeit not statistically significant. Capillary density only increased in the hypoxic high-intensity group, whereas HIF-1α increased in the hypoxic group but was independent of training intensity. LLTH also enhances exercise-induced transcription of other important genes that govern endurance performance. The effects of hypoxic training on gene transcripts were investigated in endurance runners [27]. The effects of six weeks’ training in normoxia and hypoxia (simulated altitude of 3000 m) were examined. Subjects performed their normal training program and trained under hypoxic or normoxia twice a week at the same relative intensity (second ventilatory threshold—VT2). The researchers reported a significant interaction between training-induced increases in $\dot{V}$O_2max_ and mode of training (hypoxic/normoxic), whereby the addition of short hypoxic stress to regular training induced additional transcriptional adaptations in skeletal muscle. Furthermore, they reported that PGC-1α, citrate synthase (CS), and cytochrome c oxidase subunit 4 (Cox4) mRNA significantly increased in the hypoxic group, along with HIF-1α expression. In employing the combined training effect, which integrates hypoxic training into normal training, Ponsot et al. (2006) [43] reported that the addition of two training sessions per week under hypoxic conditions (FiO_2_ = 0.145) at VT2 to their normal training enhanced mitochondrial function, improved $\dot{V}$O_2max_, $\dot{V}$O_2_ at VT2 and time to exhaustion in endurance-trained athletes after 6 weeks.

The effects of hypoxia and sprint training were [29] examined in a study that used four weeks of repeated sprint cycling training in either hypoxic (FiO_2_ = 0.146) or normoxic conditions in 50 moderately trained male cyclists. They reported an increased mRNA level of HIF-1α, carbonic anhydrase III, and monocarboxylate transporter-4, but decreased mitochondrial transcription factor A (TFAM), PGC-1α, and monocarboxylate transporter-1 (MCT1), in the hypoxic training group compared to the normoxic group, and an improved performance in cycle sprint performance. In a further study on seventeen trained cross-country skiers who trained for two weeks involving high-intensity sprint training under either hypoxic or normoxic conditions (FiO_2_ = 0.138), as with their previous work [44] also reported improved cycle sprint performance when training under hypoxic compared to normoxic conditions. In contrast to Faiss et al. [29], Robach et al. [45] compared the effects of six weeks of training in normoxia or hypoxia (FiO_2_ = 0.15) on performance in an incremental cycling test under normoxia and hypoxia conditions plus a time trial on 17 moderately trained men. They observed a lack of differences in $\dot{V}$O_2max_ changes and mitochondrial content, as assessed by cytochrome C oxidase, after training, nor any effects of hypoxia. They did, however, highlight that the lack of responses may be attributable to the training status of their subjects and that they used a lead-up training period that provided an increase in $\dot{V}$O_2max_, thereby potentially dampening the levels of increase from the training itself.

Kong et al. [46] utilized a short-term, 4-week repeated sprint interval training (RSIT) program. Subjects were randomized into a Hypoxic group (2500 m), in which participants exercised under a stable hypoxic condition (FiO_2_: 0.155, 2500 m), and a Hypoxic 2500–3400 m group, in which participants exercised under an incremental hypoxic condition (FiO_2_ was set at 0.155 initially, then decreased by 0.005 every week until the FiO_2_ dropped to 0.135). In this study, $\dot{V}$O_2peak_ did not change following 4 weeks of normoxic RSIT training when compared with the no-training controls. In contrast, the $\dot{V}$O_2peak_ of hypoxic groups was increased by 8.2% in the Hypoxic 2500 group (FiO_2_: 0.155) and 10.9% in the Hypoxic 2500–3400 group (FiO_2_: 0.155–0.135). This provides further evidence of the additional beneficial effects of hypoxia in that it seems able to compensate for the reduced training dose.

Based on the majority of results from LHTL and LLTH studies, it is reasonable to suggest that the combination of training and hypoxia provides a more potent stimulus to elicit physiological adaptations than normoxic training. Further, although some studies have suggested that responses in HIF-1α increase in hypoxia but are independent of intensity of training, most papers had used moderate to high intensity training. These findings support the role of ROS, which would presumably have higher concentrations following high-intensity work and working under hypoxic conditions. However, as outlined earlier, precisely how ROS and other metabolites modulate PGC-1α signaling and other pathways is uncertain.

## 6. Hypoxic Training in the Thoroughbred Horse

The application of the concept of hypoxic training to the Thoroughbred horse is new, and now, with purpose-built chambers that enable the air to be adjusted to whatever oxygen percentage is required, encourages its use in equine training programs (see Figure 1). The placement of these chambers over high-speed equine treadmills enable exposure to a hypoxic environment similar to that experienced at altitude but at normal barometric pressure.

The training of the equine athlete in the hypoxic chamber mimics the LLTH methods that have been employed in human studies and remains an important type of training for human endurance athletes. As the Thoroughbred horse is considered an elite athlete with a wide range of extreme physiological characteristics enabling both high anaerobic and aerobic metabolic capabilities [47], this makes their potential to respond well to hypoxic training very plausible. As previously discussed, Tyler et al. [7] showed that in horses, $\dot{V}$O_2max_ continuously increases with training, with mitochondrial volume density also increasing throughout the training, paralleling increases in $\dot{V}$O_2,max_. As outlined earlier, hypoxia triggers activation of HIF, which activates genes involved in angiogenesis, glycolysis, and potential mitochondrial biogenesis via the HIF-1α subtype of HIF. Hypoxic training could then potentially promote mitochondrial adaptations and enhance peripheral adaptations. However, although mitochondrial volume has been shown to increase in normal training, there is the potential for additional adaptative responses under hypoxic conditions. Further hypoxia may provide the potential to gain the same level of mitochondrial adaptation as per high-intensity training but without the added stress of high-speed gallops. Thus, while there is evidence to suggest that hypoxic training could promote mitochondrial adaptations, the extent of these adaptations and their practical implications require further investigation.

The physiological capacity of the Thoroughbred horse has been well documented, with $\dot{V}$O_2max_ values ranging from ~100–160 mL·kg^−1^·min^−1^ [7,9,48] which are superior to many other athletic species. These extreme capabilities are possible due to the capacity and adaptations within the respiratory and cardiovascular systems, which incorporate a large lung volume, high hemoglobin concentration, and cardiac output, as well as large muscle mass, high skeletal muscle mitochondrial density, and oxidative enzyme activity [49]. As outlined previously, the $\dot{V}$O_2max_ is generally restricted by transport or oxygen delivery supply to the mitochondria rather than by mitochondrial oxidative capacity *per se*. The major factors that limit performance capacity are all responsive to training, as outlined in human studies, and are augmented by the addition of hypoxic stimuli. Therefore, the physiological make-up of the horse makes it an ideal individual to benefit from hypoxic training.

Few studies have examined the effects of hypoxic training on Thoroughbred horses, with early studies examining physiological variables [50,51]. For instance, packed cell volume, total blood volume, red cell volume, hemoglobin, 2,3-diphosphoglycerate, blood lactate, heart rate, and plasma erythropoietin were studied, as well as speed on the track following 9 days at high altitude (3800 m). They reported increases in total red cell volume and hemoglobin concentrations, plus improved heart rate recovery and lactate recovery following the acclimatization at simulated altitude. Others supported similar improvements in cardiovascular and respiratory parameters following hypoxic training, with horses trained at high altitude having lower heart rates and higher arterial oxygen saturation during exercise compared to horses trained at sea level. However, other studies failed to find significant benefits from hypoxic training in horses. One needs, however, to be careful in comparing results as the protocols in reference to the type of training and training intensities and durations are inconsistent.

In a study by Davie et al. [52], eight horses were divided into two groups of four and trained for six weeks on an incremental moderate-intensity treadmill program. One group trained three times per week in hypoxic (15% inspired O_2,_ HT), while the other group trained in normal air (NC). Pre- and post-training, each horse completed an incremental treadmill test with heart rate and blood lactate measurements taken. Muscle biopsy samples were taken before and 24 h post-training for analysis of mRNA changes in key genes including VEGF, Peroxisome proliferator-activated receptor gamma (PPARγ), HIF-1α, PGC-1α, Cytochrome c oxidase subunit 4 (COX4), Adrenylate kinase 3 (AK3), Lactate dehydrogenase (LDH), Phosphofructokinase (PFK), Pyruvate kinase (PKm), and Superoxide dismutase 2 (SOD-2). No significant differences between the HT and NC were detected. There were no significant differences between groups for heart rate and blood lactate during the treadmill test. This hypoxia training program did not appear to modulate the resting expression of the selected mRNAs compared with normoxic training. Despite the lack of statistically significant findings, this study encourages further investigations into the signaling mechanisms modulated by a single bout of hypoxic training and protein content following long-term training programs, as protein levels do not always mimic mRNA abundance.

One investigation by Ohmura et al. [53] measured the effects of three weeks of hypoxic training on $\dot{V}$O_2max_ in five well-trained horses, during which they trained with hypoxia (15% inspired O_2_) twice a week. $\dot{V}$O_2max_ increased after 3 weeks of hypoxic training (178 ± 10 vs. 194 ± 12.3 mL O_2_ (STPD)/(kg × min), and absolute $\dot{V}$O_2max_ also increased after hypoxic training (86.6 ± 6.2 vs. 93.6 ± 6.6 l O_2_ (STPD)/min). In a further study [54], they examined the effects of high-intensity training in normobaric hypoxia on aerobic capacity and exercise performance in a randomized, crossover design with training in hypoxia (HYP; 15.0% inspired O_2_) or normoxia (NOR; 20.9% inspired O_2_) 3 days/week for 4 weeks separated by a 4-month washout period. Each training session consisted of 1 min of cantering at 7 m/s and 2 min of galloping at the speed determined to elicit $\dot{V}$O_2max_ in normoxia. They reported statistically significant differences in run time to exhaustion (HYP, +28.4%; NOR, +10.4%), $\dot{V}$O_2max_ (HYP, +12.1%; NOR, +2.6%), cardiac output (HYP, +11.3%; NOR, −1.7%), and stroke volume (SV) at exhaustion (HYP, +5.4%; NOR, −5.5%) after training in hypoxia compared to normoxia. In a subsequent follow-up study [55], whether horses trained in moderate (FIO_2_ = 16%) and mild hypoxia (FIO_2_ = 18%) demonstrated similar improvements in performance and whether the acquired training effects are maintained after 2 weeks of post-hypoxic training in normoxia was investigated. They reported that 4 weeks of training in moderate but not mild hypoxia elicits greater improvements in performance and running economy than normoxic training and that these effects are maintained for 2 weeks of post-hypoxic training in normoxia [55].

The current literature seems to suggest that training intensity may be an important factor governing the superior physiological adaptations following LLTH hypoxic training approaches in thoroughbreds favoring intensities above lactate deflection point, as a moderate intensity training program failed to find any performance or physiological benefits after hypoxic training [52].

## 7. Conclusions

Despite mixed findings, hypoxic training remains a promising area of research for improving athletic performance in the equine athlete. Further studies are needed to refine the training protocols and durations needed to elicit beneficial effects and potentially maximize physiological adaptations in the equine athlete. This may include exercise prescription performed below, around, or well-above LIP; high-intensity interval training versus continuous protocols; incline versus horizontal treadmill protocols; and various oxygen availability (i.e., O_2_%). The latter may shed light on the optimal “altitude” required to confer physiological adaptations on the horse. Researchers are encouraged to examine possible biomarkers of hypoxic training responsiveness (i.e., high responders), which could help individualize training protocols for horses. As in humans, horses may possess a unique individual capacity to respond positively to hypoxic training. Furthermore, recent evidence suggests hypoxic training has a potential role in clinical application and exercise rehabilitation, which is worth investigating in horses. Considering limb injuries are not uncommon in racehorses, low-intensity hypoxic training could be one way of maintaining aerobic fitness when high-impact running is contraindicated in the early stages of rehabilitation. Hypoxic training may also serve as a potent physiological stressor that does not rely on high stress/strain on the musculoskeletal system, which could be implemented to balance training volumes, limit the need for high-speed work on the track, and prevent orthopedic injuries. Thus, these efforts are encouraged and could significantly contribute to optimizing endurance performance and animal welfare for the equine athlete.

## Figures and Tables

**Figure 1 animals-13-02799-f001:**
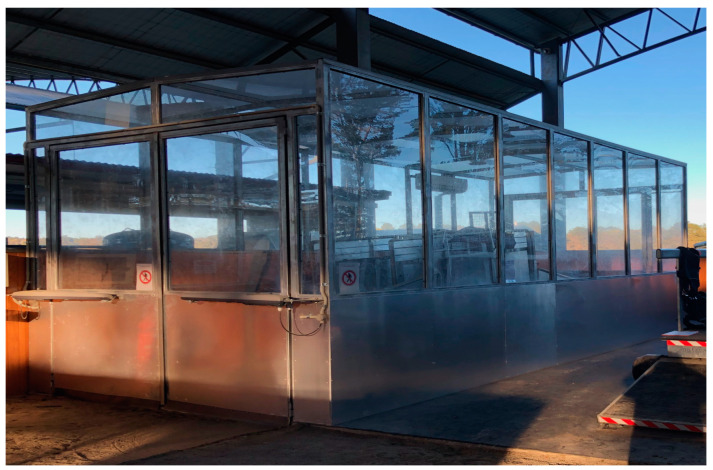
Diagram showing the structure of a hypoxic chamber with a high-speed treadmill inside.

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
