# Peer review of "The Emerging Role of Hypoxic Training for the Equine Athlete"

_animals, 2023, doi:10.3390/ani13172799_

Round 1
Reviewer 1 Report
The authors provide a good description of the state of the art about the topic (effect of training under hypoxic conditions in the horse) and discuss adaptation to training under different condition in humans, first, and horses, secondly. A complete description of the biochemical pathways is given.
First they describe knowledge in human medicine and present the opportunity of the hypoxic chambers for use in horses. A short section is dedicated to horses, compared to humans, probably due to the consistency of literature.
I would suggest the following minor revisions and accept afterwards the submission:
line 183: I would add a reference range rather than list all the number of references
references 2 and 4: pages interval is missing
reference 5 and 42: year must be in bold character
reference 14: check for consistency of reference style
Reviewer 2 Report
The authors present an overview of the role of cellular hypoxia (and the possible methodology for enhancing this stimulus) in athletic conditioning. One general statement, which should be applied throughout the manuscript, is the idea that while the phenomenon of mammalian exercise shares many of the same features across species, there are also many species-specific elements and as a result (and as the authors state early in the manuscript) findings in one species may not translate to findings in another. With this in mind and considering that the purpose of this article is specific to equine athletes, the arguments and data used to illustrate arguments and conclusions should be of equine origin whenever possible. Furthermore, given the basic premise that the more detailed the examination, the more likely one will encounter species-specific differences, greater caution regarding the use of non-equine data is warranted when discussing events at a cellular and subcellular level.
Lines 64-73: Linking genes to athletic performance – authors should rely more on existing equine data to make this point (or at least mention the existing equine data if this point is going to be needed).
Lines 75-87: The statement that VO2max is the most important factor limiting endurance performance, although supported by citations, is increasingly rejected in modern human literature in favor of critical power or similar indices. Strictly speaking, their statement that a high VO2max is an indisputable requirement for elite performance is correct, but only in the sense that VO2max is defined in large part by high cardiac output (a requirement for both maximal and sustained submaximal athletic performance). It is misleading and illogical to suggest that the capacity to consume oxygen defines the capacity for endurance performance which is performed at intensities lower than maximal aerobic capacity. The association between maximal oxygen consumption and endurance performance is largely due the fact that oxygen is the analytes used in the calculation.
Lines 97-98: Although high intensity exercise is when the ATP-PC-glycolysis system is used to the greatest extent, during these periods aerobic production of ATP still exceeds anaerobic production.
Lines 101-102: Actually, only about 20-25% of inspired oxygen is consumed. FiO2 is 20.8% (unless supplemental oxygen is being delivered), and exhaled breath contains about 15-16% O2.
Lines 114-128: Most of the points made in this paragraph can be (and should be) supported by references describing equine responses to exercise stress. References to human studies can be used to reinforce, but if the point of this article is to propose a role of hypoxia in equine athletic conditioning, then the foundation should be built with equine-specific data whenever possible.
Lines 295-296: In fact, ROS production by mitochondria has been shown to be HIGHER with high O2 compared to low O2. Whatever mechanism underlies the benefits of hypoxic training in humans, it probably isn’t due to higher levels of ROS. Fortunately, there are plenty of other signals that will modulate PGC1a signalling.
Lines 299-305: This reviewer having a great deal of experience with treadmills inside hypoxia chambers, some technical aspects should be developed by the authors – most notably, the management of metabolic wastes. A rough estimate from the picture provided by the authors is that the chamber is about 2m x 2m x 4m (16m3 or 16,000 liters). An adult Thoroughbred horse exercising at or near maximal aerobic capacity will be producing about 75 liters of CO2/minute, and thus will raise the CO2 concentration in the chamber to dangerous levels (>2%) in just a few minutes. Similarly, that same horse is producing metabolic heat at a rate of about 400 kcal/min, which will heat the air in the chamber rapidly and potentially create a thermoregulatory crisis. If these chambers are going to be used safely, the authors should describe how these potential complications are addressed.
Lines 307-323: The authors fail to address a very relevant issue when making the case for imposed hypoxia in training of horses: horses ALREADY develop pronounced peripheral hypoxia during strenuous exercise, far more than in humans. Given that it is unlikely that the capacity to respond to hypoxia is limitless, one must consider whether horses are ALREADY taking full advantage of peripheral responses to hypoxia and as a result, no further gains are likely. This could help explain the lack of myocyte-specific responses in the study of Davie et al. However, as shown in Ohmura et al, the cardiovascular system DOES respond to training during hypoxia, so describing the mechanisms underlying the increase in equine cardiac output may be more germane than equine myocellular responses, which appear to be unaffected by hypoxia.
Some minor editing is needed to correct typographical errors and awkward sentence structure
Reviewer 3 Report
Your comprehensive review provides valuable insights into the effects of hypoxic training on equine athletes. The article thoroughly covers various aspects such as the physiological adaptations induced by hypoxia, the potential benefits in enhancing endurance and performance, as well as the recommended protocols for implementing this training method.
I found your research to be well-structured, supported by strong evidence from previous studies, and presented in a clear and concise manner. Additionally, we appreciate the inclusion of practical recommendations that can be of great value to equestrians and trainers in optimizing the performance of their equine athletes.
Your contribution will undoubtedly enrich the knowledge base in the field of equine sports science.
Round 2
Reviewer 2 Report
The authors present an overview of the role of cellular hypoxia (and the possible methodology for enhancing this stimulus) in athletic conditioning. One general statement, which should be applied throughout the manuscript, is the idea that while the phenomenon of mammalian exercise shares many of the same features across species, there are also many species-specific elements and as a result (and as the authors state early in the manuscript) findings in one species may not translate to findings in another. With this in mind and considering that the purpose of this article is specific to equine athletes, the arguments and data used to illustrate arguments and conclusions should be of equine origin whenever possible. Furthermore, given the basic premise that the more detailed the examination, the more likely one will encounter species-specific differences, greater caution regarding the use of non-equine data is warranted when discussing events at a cellular and subcellular level.
The recommendation has been accepted and the statement of “findings in one species may not translate to findings in another” has been included.
Lines 64-73: Linking genes to athletic performance – authors should rely more on existing equine data to make this point (or at least mention the existing equine data if this point is going to be needed).
The recommendation has been accepted and equine papers have been referenced.
In the equine field single nucleotide polymorphisms (SNPs) in creatine kinase (CKM) and cytochrome c oxidase (COX4I2) have been linked to elite Thoroughbreds, myostatin (MSTN) gene locus has been shown to influence early skeletal muscle development and the aptitude for racing at short distances, and the mitochondrial haplogroup L3b has shown to have a negative association with elite Thoroughbreds.
Lines 75-87: The statement that VO2max is the most important factor limiting endurance performance, although supported by citations, is increasingly rejected in modern human literature in favor of critical power or similar indices. Strictly speaking, their statement that a high VO2max is an indisputable requirement for elite performance is correct, but only in the sense that VO2max is defined in large part by high cardiac output (a requirement for both maximal and sustained submaximal athletic performance). It is misleading and illogical to suggest that the capacity to consume oxygen defines the capacity for endurance performance which is performed at intensities lower than maximal aerobic capacity. The association between maximal oxygen consumption and endurance performance is largely due the fact that oxygen is the analytes used in the calculation.
A good point of discussion and something that could be debated; however, one accepts that Maximal oxygen uptake has historically been accepted as the best measure of functional limitation of the cardiovascular system, a key factor in endurance performance. Therefore, within this system one respects that oxygen is the analytes used in the system, however if one considers that the usage of oxygen (mitochondria) per se is not the limiting factor, but the transport to the mitochondria, then the transport or cardiac output, the key component of VO2max is a significant factor. It is actually the heart, capillarization of muscle, and density of mitochondria, that are key contributors to VO2max. The limiting factor per se is still not 100% conclusive due to the plethora of other factors potentially contributing.
It is likely that from a physiological standpoint, the reviewer and authors are largely in agreement: a robust cardiovascular system is essential for endurance exercise, and that in the context of the longitudinal assessment of fitness using VO2max, increases in fitness predominantly result from increased cardiovascular performance. However, the paragraph as written fails to make a clear distinction between a change in VO2max due to increased O2 delivery (a function of cardiovascular performance) and a change in capacity for O2 consumption (a function of available mitochondrial mass). This distinction becomes critical in the context of this paper’s topic and when comparing intensities of exercise that are oxygen limited (typical flatracing distances) vs those that are not (endurance racing). By placing undue emphasis on an analytical technique that merely uses O2 as a convenient indicator when discussing hypoxic training, the reader is likely to be confused as to whether the point of the hypoxic challenge is to improve capacity for O2 consumption (which thus far has not been demonstrated in horses) vs improvement in cardiovascular performance (which has been demonstrated somewhat in horses). As this technology gains greater acceptance, a review such as this will no doubt be referenced frequently. It is important to get things precisely correct.
Lines 97-98: Although high intensity exercise is when the ATP-PC-glycolysis system is used to the greatest extent, during these periods aerobic production of ATP still exceeds anaerobic production.
Yes, the balance between percentage contribution from either system is a revolving wheel determined by many factors, but as examiner states aerobic is still the major factor in most conditions.
Text should be changed to accurately reflect the relative contributions of aerobic and anaerobically-generated ATP during high intensity exercise (now Lines 102-103).
Lines 101-102: Actually, only about 20-25% of inspired oxygen is consumed. FiO2 is 20.8% (unless supplemental oxygen is being delivered), and exhaled breath contains about 15-16% O2.
Point accepted
Please edit the text to correct this statement (now Lines 106-107).
Lines 114-128: Most of the points made in this paragraph can be (and should be) supported by references describing equine responses to exercise stress. References to human studies can be used to reinforce, but if the point of this article is to propose a role of hypoxia in equine athletic conditioning, then the foundation should be built with equine-specific data whenever possible.
References have been included and statement added. “Therefore, development of the mitochondria is a key component of training, with in humans an adaptation appears to be fibre type specific however most changes result predominately from endurance-based training performed below lactate inflection point (or anaerobic threshold). It has been proposed that the magnitude of improvement in the mitochondria is influenced by the duration of the exercise or training session.
But this latter point, while widely discussed in human literature, is not supported by equine literature – specifically, the studies of Tyler et al that are referenced by the authors. In those studies, mitochondrial volume increased in BOTH Type I and Type II fibers throughout the 37-week training program, but with proportionately greater increases in mitochondrial density during the last 27 weeks of the program when training was likely entirely above the lactate inflection point (either 80% or 100% VO2max). Thus, the data available in horses would suggest that relatively high intensity training is more effective in increasing mitochondrial mass compared to low intensity training, which in the Tyler study was more effective in increasing VO2max (probably due to proportionately greater improvement in cardiovascular capacity).
Lines 295-296: In fact, ROS production by mitochondria has been shown to be HIGHER with high O2 compared to low O2. Whatever mechanism underlies the benefits of hypoxic training in humans, it probably isn’t due to higher levels of ROS. Fortunately, there are plenty of other signals that will modulate PGC1a signalling.
It is true that ROS production is high with high O2, this is one reason for the protocol which utilizes breathing high O2 following hypoxic work to provide the added ROS stimuli.
There is no mention of inclusion of hyperoxic stimulus as part of the training paradigm in the current manuscript – only the continued misleading statement that seems to suggest that the increased ROS production reported during normoxic exercise will continue to be present during hypoxic exercise. This has not been demonstrated and there is reason to believe that in fact the opposite would occur (which, as suggested by the authors in the response to reviewer, would necessitate a hyperoxic period to capture whatever adaptation is dependent on ROS production).
Lines 299-305: This reviewer having a great deal of experience with treadmills inside hypoxia chambers, some technical aspects should be developed by the authors – most notably, the management of metabolic wastes. A rough estimate from the picture provided by the authors is that the chamber is about 2m x 2m x 4m (16m3 or 16,000 liters). An adult Thoroughbred horse exercising at or near maximal aerobic capacity will be producing about 75 liters of CO2/minute, and thus will raise the CO2 concentration in the chamber to dangerous levels (>2%) in just a few minutes. Similarly, that same horse is producing metabolic heat at a rate of about 400 kcal/min, which will heat the air in the chamber rapidly and potentially create a thermoregulatory crisis. If these chambers are going to be used safely, the authors should describe how these potential complications are addressed.
The metabolic waste, principally the CO2 is dealt with via carbon dioxide sensors being placed inside chamber, that signal any major changes in CO2 levels. This is required under the occupational health and safety acts. Also, that large air conditioners and inside chamber to help regulate temperature changes. The chamber is approximately 3 m high by 3.6 m wide and 9 m long.
This is helpful, and certainly the considerably larger chamber than estimated by the reviewer will also help mitigate issues. However, CO2 sensors are just the first step in mitigating that issue – one then has to find a way of either scrubbing the CO2 or flushing it away with a new supply of hypoxic air. Ironically, the same “machine” that is producing the CO2 is also augmenting the hypoxia – in this reviewer’s experience, an aerobically-exercising athlete is a sufficiently-potent oxygen scrubber in a closed environment to rival most commercial oxygen scrubbing systems. Some description of the typical pattern of operation would be illuminating to the reader – how long can a horse realistically exercise (and at what intensity) before the system is simply unable to keep up?
Lines 307-323: The authors fail to address a very relevant issue when making the case for imposed hypoxia in training of horses: horses ALREADY develop pronounced peripheral hypoxia during strenuous exercise, far more than in humans. Given that it is unlikely that the capacity to respond to hypoxia is limitless, one must consider whether horses are ALREADY taking full advantage of peripheral responses to hypoxia and as a result, no further gains are likely. This could help explain the lack of myocyte-specific responses in the study of Davie et al. However, as shown in Ohmura et al, the cardiovascular system DOES respond to training during hypoxia, so describing the mechanisms underlying the increase in equine cardiac output may be more germane than equine myocellular responses, which appear to be unaffected by hypoxia.
There is no question over whether horses develop hypoxia during exercise. This has been well documented and a most recent paper Ohmura et al outlines that they reported hypoxia occurring during supramaximal exercise at which point arterial SaO2 had decreased to 82.5 ± 7.9%, and further that hypoxemia was also present during even submaximal exercise, in contrast to the most elite human athletes that experience these affects only during maximal exercise.
Statement by examiner 2 that (one must consider whether horses are ALREADY taking full advantage of peripheral responses to hypoxia and as a result, no further gains are likely).
There is no existing research to reject or support this statement, however, the fact that the current research shows additional increase in VO2max and performance in hypoxic trained horses suggests that this is not the case. To draw on the Davie et al study to support this is probably not ideal, as in that study the training protocol did not provide sufficient differences in physiological variables measured between groups and with a sample of four made it difficult to provide any significant differences. However, Ohmura et al did report changes in VO2max, with these changes being potentially peripheral or central based, but still highlight the increased adaptative stimuli from the hypoxia.
The reviewer agrees that the study of Ohmura et al is certainly relevant in demonstrating that hypoxic training can increase VO2max in horses. However, as previously iterated in this review, a change in VO2max can result from either increased cardiovascular performance or increased mitochondrial availability (or both), and the study of Davie et al would suggest that the latter doesn’t occur in horses during hypoxic training. Other references paint a different picture of peripheral responses to hypoxic training in humans, which begs the question of “Why would humans demonstrate peripheral responses and horses not?" The obviously scientifically-backed “guess” would be that in contrast to all but the most elite humans, horses routinely develop peripheral hypoxia during normoxic exercise. No biological cause and effect extends indefinitely, so if one is to believe all of the available published data on equine responses to hypoxic training, one must propose a reason for the lack of apparent peripheral responses. The idea that peripheral responses to hypoxia are already present during normoxic exercise in horses potentially can explain the differences that currently exist in the literature.. Of course, if the Davie et al study has shortcomings that the authors feel should temper the reliance on the data in that study (no study is ever perfect, so that is not intended to diminish the value of that effort), that argument can also be presented.
Some minor editing is needed to correct typographical errors and awkward sentence structure
Issues addressed
There remain several key statements that are confusing due to sentence structure. I'm pretty sure I know what the authors are trying to say based on the sum of the manuscript, but things could definitely be improved.
Round 3
Reviewer 2 Report
The authors' efforts to revise the manuscript are appreciated.